# Potential Factors Influencing Repeated SARS Outbreaks in China

**DOI:** 10.3390/ijerph17051633

**Published:** 2020-03-03

**Authors:** Zhong Sun, Karuppiah Thilakavathy, S. Suresh Kumar, Guozhong He, Shi V. Liu

**Affiliations:** 1Department of Biomedical Sciences, Faculty of Medicine & Health Sciences, University Putra Malaysia, UPM Serdang 43400, Selangor, Malaysia; cifer03@gmail.com (Z.S.); thilathy@upm.edu.my (K.T.); 2Genetics and Regenerative Medicine Research Group, Faculty of Medicine & Health Sciences, University Putra Malaysia, UPM Serdang 43400, Selangor, Malaysia; suresh@upm.edu.my; 3Department of Medical Microbiology and Parasitology, University Putra Malaysia, UPM Serdang 43400, Selangor, Malaysia; 4Institute of Health, Kunming Medical University, Kunming 650500, China; 5Eagle Institute of Molecular Medicine, Apex, NC 27523, USA

**Keywords:** Wuhan pneumonia, coronavirus, CoV, severe acute respiratory syndrome, SARS, COVID-19, angiotensin-converting enzyme 2, ACE2, SARS-CoV, 2019-nCoV, outbreak, epidemic, epidemiology, infection, drought, bat, green light, red light, wildlife, host, exposure, risk

## Abstract

Within last 17 years two widespread epidemics of severe acute respiratory syndrome (SARS) occurred in China, which were caused by related coronaviruses (CoVs): SARS-CoV and SARS-CoV-2. Although the origin(s) of these viruses are still unknown and their occurrences in nature are mysterious, some general patterns of their pathogenesis and epidemics are noticeable. Both viruses utilize the same receptor—angiotensin-converting enzyme 2 (ACE2)—for invading human bodies. Both epidemics occurred in cold dry winter seasons celebrated with major holidays, and started in regions where dietary consumption of wildlife is a fashion. Thus, if bats were the natural hosts of SARS-CoVs, cold temperature and low humidity in these times might provide conducive environmental conditions for prolonged viral survival in these regions concentrated with bats. The widespread existence of these bat-carried or -released viruses might have an easier time in breaking through human defenses when harsh winter makes human bodies more vulnerable. Once succeeding in making some initial human infections, spreading of the disease was made convenient with increased social gathering and holiday travel. These natural and social factors influenced the general progression and trajectory of the SARS epidemiology. However, some unique factors might also contribute to the origination of SARS in Wuhan. These factors are discussed in different scenarios in order to promote more research for achieving final validation.

## 1. Introduction

Since 2002, two epidemics of severe acute respiratory syndrome (SARS) have originated from China, one in late 2002 and the other in late 2019. The etiological agents of these epidemics have been confirmed as a new subset of coronaviruses (CoVs), namely, SARS-CoV and SARS-CoV-2 (Figure 1), respectively, for the 2002 and the 2019 SARS epidemics [1].

CoVs are named for their crown-like spikes on the viral surface. They are classified into four main sub-groupings known as alpha, beta, gamma, and delta. Before the emergence of SARS-CoV, four CoVs were known as human coronaviruses (HCoVs), i.e., CoVs capable of infecting human beings. These four HCoVs cause a “common cold” and include HCoV-229E and HCoV-NL63 of the alpha group and HCoV-OC43 and HCoV-HKU1 of the beta group [2]. Since the discovery of SARS-CoV causing SARS in China in 2002 [2], another HCoV was identified in 2012 as MERS-CoV, causing Middle East respiratory syndrome (MERS) [3].

SARS-CoV differs from MERS-CoV because it uses angiotensin-converting enzyme 2 (ACE2) as a receptor for binding to human cells [4]. In contrast, MERS-CoV uses dipeptidyl peptidase 4 (DPP4) as a receptor for infecting human cells [5]. Phylogenetically, SARS-CoV and MERS-CoV are distinct and both are distant from other CoVs, including HCoVs.

The recent outbreak of “Wuhan pneumonia” in late 2019 in central China has been linked with a new CoV formally identified as SARS-CoV-2. SARS-CoV-2 is not only phylogenetically closely related with SARS-CoV, an etiological agent of SARS, but also uses a same receptor, ACE2, as SARS-CoV does. Thus, even though “Wuhan pneumonia” has been called with various other disease names such as “new coronavirus pneumonia (NCP)” and now as “coronavirus disease 2019 (COVID-19)”, we feel that it may be more appropriate to refer to “Wuhan pneumonia” as “SARS-2” and the previous SARS as “SARS-1” if necessary. The etiological agent for “Wuhan pneumonia” has been changed from “2019-nCoV” to “SARS-CoV-2”. A further change of “COVID-19” into “SARS-2” is logical and reasonable for streamlining taxonomy between disease agent and disease. In this mini-review, we evaluate natural and social factors influencing both 2002 and 2019 SARSs in order to understand some common epidemiological features that may be beneficial for controlling the ongoing epidemic and also for preventing future outbreak. This comprehensive knowledge is also helpful for searching the origin(s) of the viruses and for elucidating their initial occurrence(s).

## 2. Common Epidemiological Features for SARS-1 and SARS-2

It is amazing that, within a short time span of less than 17 years, two similar epidemic outbreaks occurred in China: SARS-1 in 2002 and SARS-2 in 2019. Although identification of viral origin(s) is very critical for understanding these epidemics, a study comparing a wide variety of natural and social factors potentially influencing the progression and the trajectory of these epidemics is also important. Through a comparative analysis of environmental factors and human activities in these two serious public health events, we wish to find some common ground for the occurrence of SARS-1 and SARS-2.

### 2.1. Environmental Factors

SARS-1 broke out in Foshan, Guangdong Province, in November 2002 [6]. SARS-2 started in Wuhan in Hubei Province no later than early December 2019 [7]. In China, November and December are winter months, and are the coldest months of the year in these two locations [8,9]. Cold temperature usually provides a conducive environmental condition for virus survival. In addition to this, we also noticed that severe drought occurred in both locations at the times of the outbreaks. The annual rainfall in Foshan in December 2002 nearly reached 0 mm [10]. In fact, drought occurred in the whole of Guangdong Province that year, causing more than 1300 reservoirs drying up and 286,000 hectares of farmland suffering drought [11]. Coincidentally, Wuhan also suffered its worst drought in nearly 40 years, with precipitation of only 5.5 mm in December 2019 [12,13]. These drought conditions were rare for both locations as their average annual precipitations are greater than 1100 mm [8,9], which are higher than the global average annual rainfall of 990 mm, of which 715 mm is over land [14]. Cold, dry conditions are more conducive than cold conditions alone for virus survival [15,16]. During the cold winter, air-dried virus particles are a dangerous form of virus, which survives for a long period of time in airflows [17].

Besides providing conducive conditions for virus survival and spreading, winter cold conditions also damper humans’ innate immunity. Cold temperatures cause reduced blood supply and thus the decreased provision of immune cells to the nasal mucosa. Low humidity can reduce the capacity of cilia cells in the airway to remove virus particles and secrete mucus as well as repair airway cells. In addition, human cells release signal proteins after viral infection to alert neighboring cells to consider the danger of virus invasion. However, in low-humidity environments, this innate immune defense system is impaired [18]. More seriously, low humidity can cause nasal mucus to become dry; nasal cavity lining to become fragile, or even ruptured; and make the entire upper respiratory tract vulnerable to virus invasion [19].

The environmental situation of another coronavirus outbreak also seems to support the above-mentioned theory. MERS-CoV was first detected in a patient living in Jeddah, Saudi Arabia, in June of 2012 [20]. The annual rainfall in Jeddah is low at 61mm, and there was no rain at all in June of that year in Jeddah [21]. Therefore, relative to temperature, low humidity seems to be a more critical environmental factor influencing outbreak of human coronavirus disease.

### 2.2. Natural Host

For both SARS outbreaks, bat was suspected as a natural host for SARS-CoVs. It was claimed that SARS-CoV virus originated from horseshoe bats in a cave of Yunnan Province [22].

In 2005, SARS-like COVs (SL-CoVs) were found in wild Chinese horseshoe bats (*Rhinolophus sinicus*) collected from a cave in Yunnan Province of China [22]. In 2013, live SL-CoV was isolated from Vero E6 cells incubated in bat feces [23]. The isolated virus showed more than 95% genome sequence identity with human and civet SARS-CoVs. SL-CoV possesses the ability to infiltrate cells using its S protein to combine with ACE2 receptors [24]. This observation indicated that SARS-CoV originated from Chinese horseshoe bats and that SL-CoV isolated from bats poses a potential threat to humans without the involvement of any intermediate hosts. Between 2015 and 2017, 334 bats were collected from Zhoushan city, Zhejiang Province, China. A total of 26.65% of those bats were detected as having a conserved coronaviral protein RNA-dependent RNA polymerase (RdRp). Full genomic analyses of two SL-CoVs (bat-SL-CoV ZC45 and bat-SL-CoV ZXC21) showed 81% nucleotide identity with human/civet SARS CoVs. These viruses reproduced and caused disease in suckling rats, with virus-like particles being observed in the brains of suckling rats by electron microscopy [25].

Thus, prior to 2018, bats collected in some areas of China have been shown to carry CoVs capable of directly infecting humans.

A recent study showed that SARS-CoV-2 has 96% homology at the whole genome level with bat coronavirus. Pairwise protein sequence analysis of seven conserved non-structural proteins showed that this virus belongs to the species of SARS-CoV [26]. In phylogenetic analysis, SARS-CoV and SARS-CoV-2 not only share a common ancestor, but also have an amino acid identity of 82.3% [27,28,29].

### 2.3. Intermediate Hosts

Viruses often require intermediate hosts before transmitting from bats to humans. For example, the intermediate host of Nipah virus is pig, and the intermediate host of MERS-CoV is camel [30,31]. During SARS-1 outbreak, civet was initially considered as a natural host for SARS-CoV [31]. Later it was redefined as an intermediate host after bats were claimed as the natural hosts for SARS-CoV. In addition to civet, researchers also found SARS-CoV from domestic cat, red fox, Lesser rice field rat, goose, Chinese ferret-badger, and wild boar in Guangdong’s seafood market. It was believed that the virus was transmitted to civet from Yunnan horseshoe bats, and civet cats carrying the virus were transported to Guangdong, which led to SARS-CoV infection on humans and SARS outbreak in Guangdong [32].

Currently, some intermediate hosts have been suspected for SARS-COV-2. A study showed that SARS-COV-2 has the same codon usage bias as shown for snakes. Therefore, snake may be the intermediate host for SARS-COV-2 [33]. However, David Robertson, a virologist from the University of Glasgow, United Kingdom, stated, “Nothing supports the invasion of snakes.” At the same time, Paulo Eduardo Brandão, a virologist from the University of St. Paul, also said, “There is no evidence that snakes can be infected by this new coronavirus and act as hosts” [34]. A study on the genome sequence of diseased pangolins smuggled from Malaysia to China found that pangolins carry coronavirus, suggesting that pangolins may be intermediate hosts for SARS-COV-2 [35]. Pangolins seized in anti-smuggling operations in Guangxi and Guangdong of southern China were detected with multiple CoV linages with 85.5–92.4% genome sequence similarity to those of SARS-CoV-2 [36]. More interestingly, CoVs collected from caged pangolin obtained from an unspecified research organization showed over 99% genome sequence identity to those of SARS-COV-2 [37]. Meanwhile, Nanshan Zhong, the leader of the SARS-COV-2 virus treatment expert group, predicted the intermediate host of SARS-COV-2 to be bamboo rat [38] on the basis of the animal distribution in Zhoushan, which is not only the natural habitat of bat-sl-CoVzc45-carrying bats, but also the natural habitat of cobra, bamboo rat, and pangolin [39,40,41].

### 2.4. Ultimate Host

Before viruses in wildlife make a jump to infect human beings, they usually accumulate a series of mutations in their viral genomes [42] and invade human beings as a result of human occupation of their normal ecosystem, as exemplified with a story of initial human infection by HIV carried by chimpanzees in rainforests of West Africa [43,44].

At the outset, SARS-COVs might have a species barrier before it can be transmitted to humans. However, due to human activities, the virus has expanded its host of infection. It was found that the SARS-COV responsible for SARS-1 in 2002 existed in civet [32]. Viruses phylogenetically similar to SARS-COV-2 in genome sequence have now being detected in wild bats [26], snakes [33], and pangolins [35,36,37,45].

Thus, humans might become unfortunate hosts for SARS-CoVs as a result of some inappropriate interactions with wildlife and thus exposure to unfriendly viruses (Figure 2).

## 3. Potential Outbreak Scenarios for SARS-2

Having identified some relevant natural and social factors common for affecting both SARS epidemics, it is also necessary to discuss if variations in these factors contributed to the unique outbreak of SARS-2 in Wuhan. Because many factors confounding the SARS-2 epidemic are still unknown, we herein discuss SARS-2 outbreak in Wuhan (Figure 3) under different scenarios.

### 3.1. Single Outbreak Site and Single Source of Virus

An early guess and also a dominant view expressed in published reports assumes that SARS-2 outbreak started from a single site in Wuhan, namely, Huanan Seafood Market [46]. However, the only source of bats that have been publicly identified as carrying virus phylogenetically close to SARS-CoV-2 is far away from Wuhan in Zhoushan, Zhejiang. Zhoushan is also one of the largest breeding bases in Zhejiang for bamboo rat, which is suspected as one of the intermediate hosts for SARS-CoV [38,47].

Thus, in order for these bats and/or rats to pass the virus to humans, they must have first been able to migrate or be moved to Wuhan and also must have carried viruses that actually achieved mutations for affording the capability of infecting human beings.

Bats have an ability to migrate more than 1000 kilometers and tend to fly to insect-rich areas [48]. Abundant insects are often found in wildlife market areas due to their selling of various animals. Animal carcasses also make these places suitable habitats for bats. Bats are also attracted to artificial green lights and tend to gather around green light areas [49].

In agreement with these natural characteristics, bats have been found to inhabit locations near Yangtze River Bridge, which has rows of green lights that are tuned on for all of the night-time. Incidentally, Huanan Seafood Market is only 20 minutes away from this bridge. Bats gathered near the Yangtze River Bridge might have released the virus and even infected intermediate hosts for some time. The cold and dry winter helped viruses to survive in the environment and eventually found some ways to cross the species barrier, a phenomenon known as “viral chatter” [50]. The increased vulnerability of human beings in winter time and the increased human exposure to wild animals during holidays made infection to SARS-COV-2 more likely.

At present, there is no evidence to prove the source of bamboo rats in Huanan Seafood Market. Therefore, there are two possible places for bamboo rat be infected with SARS-COV-2.

The first site might be the bat habitat in Zhoushan. Due to the promotion of bamboo rat breeding by Huanong Brothers in 2018, the amount of bamboo rat breeding and market demand increased significantly [51]. Since the market demand increases, the new bamboo rat breeding base may not be far from the local habitat of SARS-COV-2-carrying bats. The model of SARS-COV-2 transmission, similar to Nipah virus, is that farms are built around bat habitats, causing bats to pass the virus to animals through saliva, urine, and feces [30]. At the same time, because Zhejiang is a natural habitat for bamboo rats, it is possible that some farms directly introduced wild bamboo rats, which were already infected with SARS-COV-2 virus. For the above reasons, the bamboo rats carrying SARS-COV-2 virus were transported from the infected place to the incident site in the same way that civets spread SARS-CoV [32].

The second site is Wuhan, the place of the SARS-COV-2 outbreak, and it is also the end point for some bat migration. Zhengli Shi’s team from Wuhan Institute of Virology, Chinese Academy of Sciences, isolated a live SARS-like strain in the feces of horseshoe bats [23]. This suggests that the way the bats spread the virus is not only via direct contact, but also through feces. Therefore, when bats carrying SARS-COV-2 virus forage at Huanan Seafood Market, they may pass the virus directly or indirectly to intermediate hosts.

However, to confirm this scenario, it is necessary to find wild bats in Wuhan and its neighboring areas that carry CoVs identical to those isolated from various SARS-2 patients. It is also necessary to find a mechanism for the very quick outbreak in such a wide area by a natural source of SARS-CoV-2.

### 3.2. Multiple Outbreak Sites and Multiple Sources of Viruses

Epidemiological investigations showed that 13 of the first 41 patients diagnosed with SARS-CoV-2 had nothing to do with Huanan Seafood Market [45]. Another survey of SARS-2 found that no bats were on sale in Huanan Seafood Market [52].

With so many bats concentrated into a local area, the spreading of viruses by bats might be much wider than just being restricted to one wildlife trading place such as the Huanan Seafood Market. The viruses might have lived in this big “incubation bed” for some time and achieved some mutations before jumping on to the final hosts—human beings.

A study on horseshoe bats in Hong Kong and Guangdong showed that the viruses carried by horseshoe bats in these two places are different. However, some horseshoe bats were found to carry two viruses after mating and foraging activity. This indicates that horseshoe bats not only have the ability to migrate, but also the ability to promote the spread of virus within the same roost and from roost to roost. In addition, sequencing the entire genome of virus carried by bats in multiple regions revealed frequent recombination among different strains. For example, civet SARSr-CoV SZ3 recombination was detected between SARSr-Rh-BatCoV Rp3 from Guangxi, China, and Rf1 from Hubei, China [53]. Therefore, there is a possibility that SARS-CoV-2 spread from Zhoushan to Wuhan due to bat migration.

It turned out that bats are not only attracted by green lights but also red lights [54]. Along the Yangtze River there are also huge bridges decorated with a massive number of red lights. Thus, bats migrating along the Yangtze River might be attracted by these red lights and be relocated nearby. Wuhan might be a new habitation site for a massive number of bats. These bats, coming from different locations, might carry different virus strains. The separate evolution and the recombination of these viruses might lead to the creation of various SARS-CoVs capable of cross-species transmission and ultimate infection of human beings.

### 3.3. Multiple Outbreak Sites and Single Unique Source of Virus

Many observations have shown the outbreak of SARS-2 actually started from multiple sites, instead of just a single site, as originally reported [27,52,53,55]. In evaluating the epidemiological patterns of SARS-2 within Wuhan, surrounding Wuhan, and remote from Wuhan, it appears that the incidences of SARS-2 have some distinct patterns. Although the remotely occurring SARS-2 usually have a human–human linkage and can be traced to a single source of infection, some Wuhan cases and the surrounding cases in Hubei Province still lack reliable sources of infection. Amazingly, most of the SARS-2 patients can be traced to a single unique etiological agent, SARS-CoV-2. How could this likely single source of virus quickly infect so many people in such large geographic area? This is a question that is difficult to answer now, but must be answered in future.

## 4. Recommendations for SARS Control and Prevention

### 4.1. Control Measures for the Ongoing SARS-2 Epidemic

Although the origins and the occurrences of SARS-CoV-2 are both unclear, the control measures for the current epidemic should focus on immediate cut-off of transmission of the disease and through disinfection of infected locations. Quarantine of patients (both confirmed and suspected), isolation of susceptible population, and protection of high-risk professions are necessary measures for reducing exposure to the viruses and eliminating the risk of getting infected by the viruses. At the same time, infected locations must be adequately disinfected. Areas that will be open to the public should be carefully surveilled for the existence of SARS-CoV-2 and be cleaned of the virus if it is found. Modern communication methods should be effectively used for passing reliable information on the epidemic status, the treatment measures, and the self-protection skills, among others. As a matter of fact, if fine-tuned and highly-effective internet control for “public opinions” can be turned into beneficial use of monitoring the “epidemic situation”, fighting against an even larger outbreak of any infection would be much easier and cost-effective.

### 4.2. Prevention Strategies for Potential SARS in Future

SARS-CoV-2 has entered human communities, and eliminating virus from human bodies does not means its eradication in nature. The risk of SARS-CoV-2 infection will remain for a long time. Thus, adequate cautions must be taken for safe-guarding against future outbreaks of SARS. The prevention can be achieved by implementing a multi-facet system that considers both natural and social aspects of the SARS epidemiology discussed earlier. For example, regular surveillance of viral status in nature should be carried out to monitor the variation/evolution and abundance/localization of the virus. This information may be served as an early warning and used for preparation of potential vaccines. The government should issue laws and policies to tighten protection of wildlife and prohibit consumption of wild animals. A grass-roots and transparent reporting system should be established and put into public use for reporting any case of confirmed or suspected human infection. The disease-reporting system should be organically synchronized with the meteorological system so that adverse environmental conditions conducive for viral infection on human beings can be forecasted and macro-scale preparations can be made in case an emergency occurs. Finally, but not lastly, in developing human society including building massive constructions for residence and transportation, potential ecological impact on wildlife and possible consequences of breaking natural balance of the ecosystems should be carefully evaluated.

## 5. Conclusions

This mini-review evaluated the common epidemiological patterns of both SARS epidemics in China and identified cold, dry winter as a common environmental condition conducive for SARS virus infection to human beings. Thus, meteorological information should be integrated into future forecast of potential outbreak of new SARS. The identification of bats as very likely natural hosts for SARS-CoVs and consideration of some other wild animals as potential intermediate hosts leads to a prevention requirement of protecting natural ecosystem and prohibiting consumption of wildlife. The presentation of different scenarios of SARS outbreaks points to some urgency in identifying the true origin(s) of SARS-CoVs and establishing more comprehensive anti-infection measures that will resist any kind of viral assault.

## Figures and Tables

**Figure 1 ijerph-17-01633-f001:**
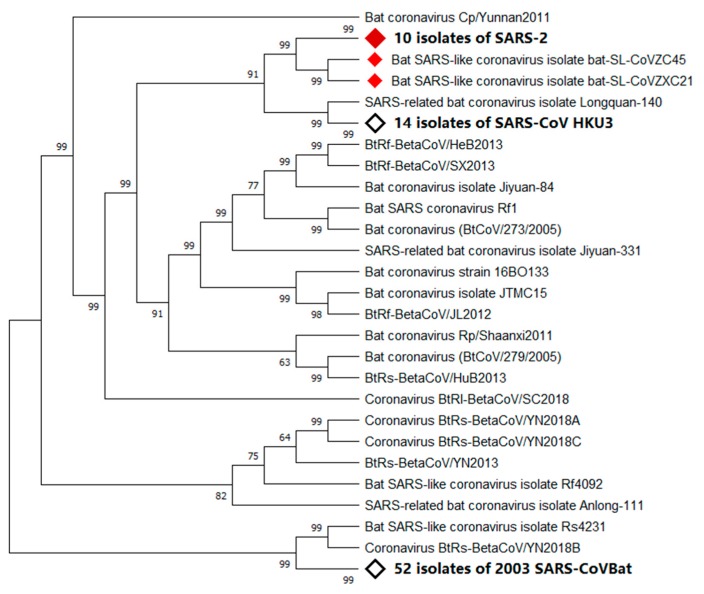
Phylogenetic analysis of virus isolated from severe acute respiratory syndrome (SARS)-2 patients. Sequence of Wuhan seafood market pneumonia virus isolate Wuhan-Hu-1 was used for comparing with whole genome sequence database from National Center for Biotechnology Information (NCBI) by using Basic Local Alignment Search Tool (BLAST). MAFF (AIST) was used to align the first 100 matching sequences. Phylogenetic trees were constructed by using MEGA X through neighbor-joining (NJ) methods. According to the phylogenetic tree, SARS-2, bat SARS-like coronavirus isolate bat-SARS-like coronavirus (SL-CoV) ZC45, and bat SARS-like coronavirus isolate bat-SL-CoVZXC21 share a common ancestor.

**Figure 2 ijerph-17-01633-f002:**
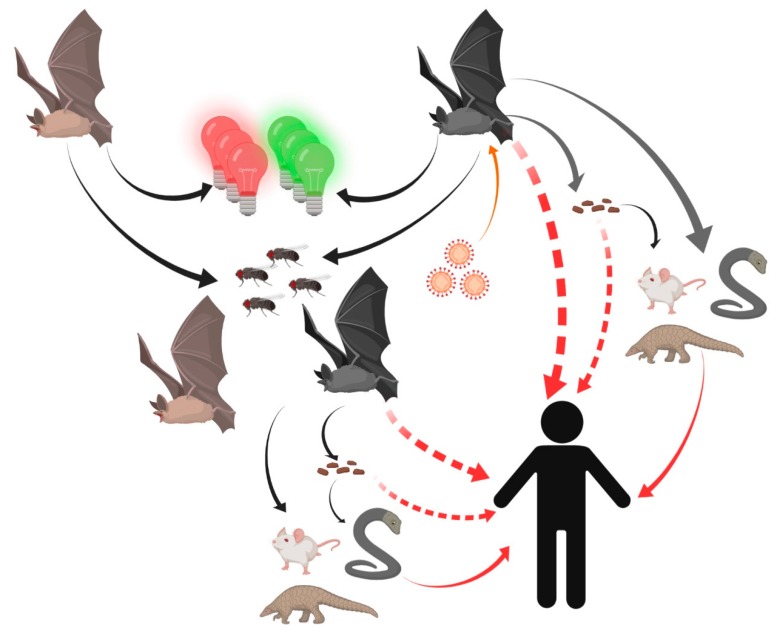
Potential transmission routes for SARS-CoV-2 to humans. Bats carrying SARS-CoV-2 were attracted by green or red lights and settled into insect-rich areas. SARS-CoV-2 transmitted to humans directly or spread to intermediate hosts such as bamboo rats, snakes, and pangolins through bats’ saliva, urine, and feces. Intermediate hosts transmitted SARS-CoV-2 to humans.

**Figure 3 ijerph-17-01633-f003:**
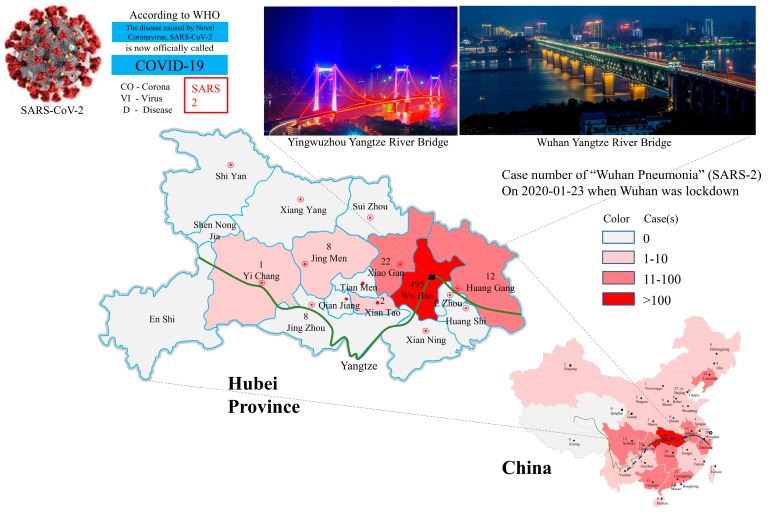
SARS-2 outbreak in Wuhan and its spreading into various places in China with a heavy impact on neighboring areas in Hubei Province. The cases shown on maps reflect a snapshot of the epidemic on the day when Wuhan was placed under a lockdown. Known and potential migration routes for bats are shown with solid and broken lines, respectively, on the China map. The Yangtze River is shown with a green line in both maps for China and Hubei Province.

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
