# Peer review of "Potential Factors Influencing Repeated SARS Outbreaks in China"

_ijerph, 2020, doi:10.3390/ijerph17051633_

Round 1

Reviewer 1 Report

Sun et al reviewed the possible factors for the outbreak of 2019-nCoV in Wuhan, It's a very interesting review and provided valuable information for figure out the transmission routine of the virus

Page 2, line 73, page 3 line 120, the Institution for Zhengli Shi, is Wuhan Institute of virology, Chinese Academy of Sciences

Author Response

Point :

Sun et al reviewed the possible factors for the outbreak of 2019-nCoV in Wuhan, It's a very interesting review and provided valuable information for figure out the transmission routine of the virus.

Page 2, line 73, page 3 line 120, the Institution for Zhengli Shi, is Wuhan Institute of virology, Chinese Academy of Sciences. 

Response:

Thank you for your affirmation and suggestions for our manuscript. We have revised in new manuscript according to your suggestions:

 (Page 2, 82. Page 4, 159)

Zhengli Shi's team from Wuhan Institute of virology, Chinese Academy of Sciences

Meanwhile, we have improved the logic and readability for new manuscript and apply for English editing service from MDPI.

Please allow us to submit new version of manuscript. Thank you

Reviewer 2 Report

This is an interesting and timely manuscript reviewing causes for the emergence of a novel strain of Coronavirus in China.  Although new experimental data is not presented in the manuscript, the authors discuss risk factors and environmental pathogenesis leading to the emergence of 2019-nCoV.  Various aspects are discussed including effects of weather, human alteration of the environment, and dietary practices.  The authors also provide suggestions to limit future novel virus outbreaks including monitoring of drought stricken areas, changing of outdoor lighting colors, and animal screening prior to transportation to market. 

This reviewer has no significant concerns regarding the manuscript.  Minor English grammatical corrections are necessary.

Author Response

Point:

This is an interesting and timely manuscript reviewing causes for the emergence of a novel strain of Coronavirus in China.  Although new experimental data is not presented in the manuscript, the authors discuss risk factors and environmental pathogenesis leading to the emergence of 2019-nCoV.  Various aspects are discussed including effects of weather, human alteration of the environment, and dietary practices.  The authors also provide suggestions to limit future novel virus outbreaks including monitoring of drought stricken areas, changing of outdoor lighting colors, and animal screening prior to transportation to market.

This reviewer has no significant concerns regarding the manuscript.  Minor English grammatical corrections are necessary. 

Response 1:

Thank you for taking time to review our manuscript. Your affirmation gave us the confidence to improve manuscript.

Meanwhile, We also apply for English editing service from MDPI.

Please allow us to submit new version of manuscript. Thank you

Reviewer 3 Report

In their review the authors search for clues of possible origin of the 2019-nCoV infections. Based on my reading, I conclude that the review is not well-ground and it is risky to consider it as a candidate for publication. Even if I was able to catch some discrepancies, I believe there are some others. I assume that some discrepancies are due to English, but sometimes the authors just make some suggestions sound as they are the proven facts. I regard this as a dangerous practice. I do not recommend it for publication in IJERPH.   Major concerns: - L72: As far as I know, it has not been officially confirmed that the origin of nCoV-infection is in bat species. It is most likely possible, but not 100%. I recommend to change the first sentence to reflect this fact. - L79-92: I don't think that the focus is correct in those two paragraph. First, bats can be "attracted" to the markets, because of humans: they can be sold on the market or some contaminated agricultural products can be brought to the market. - L115: Similar to my first remark: at the moment, 2019-nCoV carrying bats have not been identified as far as I know.  - L117: The phrase "bamboo rats, which were already infected with 2019-nCoV virus" contradicts to the previous statement. As L108 reads, the leader of the 2019-nCoV predicted, but it is only the hypothesis of that leader, while L117 mistakenly states this as a fact. The authors need to change the L117, as to state that the infection with 2019-nCoV is probable or hypothetical. - L125: Do the authors imply the first patient indicated as the one with illness onset on 1 December 2019 indicated in Lancet paper (see Fig 1 in Huang et al 2020 doi:10.1016/S0140-6736(20)30183-5)? Or is it a hypothesis? It is important to be clarified and the necessary reference should be cited. - L133: It is HIV, not AIDS virus. As far as I remember, the timing was around 1920s, and again it was stated that it was a probably jump from SIV infection in chimpanzees. The authors should put the relevant references and also be careful with their statements. The whole paragraph and L134-135 in particular sound quite speculative and reflect more of the authors' personal point of view. - Figure 1: It is impossible to read any of the labels and understand anything from the Figure.   Minor: - L20: What is "its onset"? The onset of 2019-nCov or of other outbreaks of coronavirus? I think there is a small confusion about English.  - L54: I would remove "only", because 715 mm is not much less than 990 mm. - L60-70: It is a nice paragraph. - L103-109: If talking about news coverage, there were some recent news on connection of nCoV to pangolin. Maybe the authors could indicate this also, but I am not sure how trustful they were. - L110: I do not understand the meaning of "the origin of bamboo rats in Huanan Seafood Market". Could the authors re-phrase it? 

Author Response

First of all, we thank this reviewer for carefully evaluate our manuscript and give many reasonable criticisms.  Many of the points raised by the reviewer reflect the uncertainty of defining the origin(s) of the virus(es) causing current epidemic started from Wuhan, China.  We have addressed this general concern on our original manuscript by eliminating all confirmative statements attributing the viral origin(s).  Instead we place all reported sources of viruses only as potential origins for the epidemics as all these viruses examined so far are just phylogenetically related to the actual virus causing the disease.

Besides this cross-board change in our attitude towards confirmation of viral origin(s) our manuscript also breaks from “A review on the factors contributing to COVID-19 virus outbreaks in Wuhan” (the original title) and changes into evaluating “Potential factors influencing repeated SARS outbreaks in China” (the current title).  By setting a more comprehensive framework in this significantly revised version we are now dissecting the epidemiology of two similar epidemics (SARS and COVID-19 as currently called or SARS-1 and SARS-2 as we proposed) in order to find some common features.  We also present some different scenarios for the outbreak of COVID-19/SARS-2 in order to promote more reasonings and researches so that more defined understanding of the true origin(s) of the SARS-CoV-2 can be obtained.

With this brief introduction to our overhaul on original manuscript being presented we will answer following specific questions in reflection of our modified text.

Point 1

- L72: As far as I know, it has not been officially confirmed that the origin of nCoV-infection is in bat species. It is most likely possible, but not 100%. I recommend to change the first sentence to reflect this fact.

Response 1:

This criticism is correct.  In the section discussion “Natural host” which focus on bats we objectively present findings on SARS-CoV and SARS-CoV-2 and only state that some CoVs carries by bats were shown as “capable of directly infecting humans”  But this possibility needs validation as we stated in the “Scenarios” section 3.1 “wild bats that carry CoVs identical to those isolated from various SARS-2 patients” need to be found.

(Page 4,5 166-209, especially 206-209)

Point 2

- L79-92: I don't think that the focus is correct in those two paragraph. First, bats can be "attracted" to the markets, because of humans: they can be sold on the market or some contaminated agricultural products can be brought to the market.

Response 2:

This confusion in focus has been addressed with separate discussions on bats as natural host in section 2.2 (Page 3, 101-122) and as potential source of infection for SARS-2 in section 3 (Page 3-5, 161-243). There is evidence that there are no bats on the market (Page 5, 212-213). Thus, as a natural host bat can be attracted to some markets and contaminate some agricultural products.  Thus, the potential of bats causing SARS exists if they carry SARS-CoV.  However, whether SARS-2 was caused by bats in Wuhan Huanan Seafood Market, there is no direct evidence so far. Thus, we treat bats only as potential natural hosts for spreading SARDS-CoV-2 in various scenarios.

Point 3

- L115: Similar to my first remark: at the moment, 2019-nCoV carrying bats have not been identified as far as I know.

Response 3:

Agree.  We corrected our manuscript to reflect this (see answer to point 1).

Point 4

- L117: The phrase "bamboo rats, which were already infected with 2019-nCoV virus" contradicts to the previous statement. As L108 reads, the leader of the 2019-nCoV predicted, but it is only the hypothesis of that leader, while L117 mistakenly states this as a fact. The authors need to change the L117, as to state that the infection with 2019-nCoV is probable or hypothetical.

Response 4:

We have greatly expanded the section on “Intermediate host” (Section 2.3). (Page 3, 123-148) Bamboo rats are only one type of suspected intermediate host.  We also eliminated the conflicting presentations by re-writing a new section in the Scenario 3.1 to objectively citing report on bamboo rats for their potential role as intermediate hosts. (Page 4,5, 166-209)

Point 5

- L125: Do the authors imply the first patient indicated as the one with illness onset on 1 December 2019 indicated in Lancet paper (see Fig 1 in Huang et al 2020 doi:10.1016/S0140-6736(20)30183-5)? Or is it a hypothesis? It is important to be clarified and the necessary reference should be cited.

Response 5:

We now present claims for a single origin site and for multiple origin sites as two potential scenarios.  Detailed descriptions with related references are in Sections 3.1 and 3.2, respectively. (Page 4,5, 166-233)

Point 6

- L133: It is HIV, not AIDS virus. As far as I remember, the timing was around 1920s, and again it was stated that it was a probably jump from SIV infection in chimpanzees. The authors should put the relevant references and also be careful with their statements. The whole paragraph and L134-135 in particular sound quite speculative and reflect more of the authors' personal point of view.

Response 6:

Thank you for your correction.

In fact, this paragraph comes from a book " Vanishing Borders: Protecting the Planet in the Age of Globalization" rather than our subjective assumptions. And we found some article on the origin of HIV mentioned that according to calculations HIV originated between 1910-1940.  (Page 4, 150-153)

We have added relevant citations.

Point 6

- Figure 1: It is impossible to read any of the labels and understand anything from the Figure.   Minor:

Response 6:

Thank you for your suggestion. We have revised Figure 1. (Page 6)

Point 7

- L20: What is "its onset"? The onset of 2019-nCov or of other outbreaks of coronavirus? I think there is a small confusion about English.

Response 7:

This issue does not exist as the section is re-written entirely without this unclear wording.

Point 8

- L54: I would remove "only", because 715 mm is not much less than 990 mm.

Response 8:

Thanks for point this out. We rephrased this as:

These drought conditions were rare for both locations as their average annual precipitations are greater than 1100 mm [8,9], which are higher than the global average annual rainfall of 990 mm, of which 715 mm is over land [13]. (Page 2, 82-85)

Point 9

- L60-70: It is a nice paragraph.

Response 9:

Thank you.

Point 10

- L103-109: If talking about news coverage, there were some recent news on connection of nCoV to pangolin. Maybe the authors could indicate this also, but I am not sure how trustful they were.

Response 10:

We added some recent publications on detecting in pangolins viruses claimed of related with SARS-CoV-2. (Page 3, 138-144)

Point 11:

- L110: I do not understand the meaning of "the origin of bamboo rats in Huanan Seafood Market". Could the authors re-phrase it? 

Response 11:

We change “origin” to “source”

At present, there is no evidence to prove the source of bamboo rats in Huanan Seafood Market. (Page4, 188,189)

Reviewer 4 Report

Thanks a lot for providing me with the unique opportunity of serving as a reviewer for this manuscript, submitted to IJERPH for potential consideration and publication. 

I have read it with great interest and after carefully assessing it I have the following suggestions and recommendations, that could improve the quality and the meaning of the manuscript. 

1) Generally speaking, English is poor and should be significantly improved. Authors should work to improve the and enhance the readability of the manuscript and its flow. 

2) Since the topic is timely and fast evolving, please update the manuscript: nCov-2019 has now a name, which is Covid-19. nCov-2019 was only a transitory designation.

3) Introduction should be considerably enriched.

4) Figures are of poor quality and should be provided in a higher resolution format. 

5) The purpose of comparing coronaviruses-outbreak is interesting, I think authors should add the MERS. 

6) The type of article is confusing: authors speak of analyses, but the format is a review. Please clarify. 

7) Concerning the phylogenetic tree analysis, how authors performed it, using which software and method/technique. Data are missing and, as such, not well reproducible. Authors speak only of BLAST but should provide more details. 

8) There are other articles and studies which should be quoted. 

9) Authors should explain the novelty of their study and how can advance and significantly contribute to the field. 

Author Response

First of all, thank you for your comments and suggestions that allowed us to greatly improve the quality of the manuscript. We agree with all your comments, and we addressed them in the manuscript accordingly.

Point 1

Generally speaking, English is poor and should be significantly improved. Authors should work to improve the and enhance the readability of the manuscript and its flow.

Response 1:

Thank you for taking time to read our manuscript, we apologize for the problems caused by the quality of English.

Now a senior scientist working in US for over 30 years and published many excellent papers in top journals including Science has joint us in re-writing this manuscript.  We hope this will greatly overcoming our shortcoming in using English.

Point 2

Since the topic is timely and fast evolving, please update the manuscript: nCov-2019 has now a name, which is Covid-19. nCov-2019 was only a transitory designation.

Response 2:

It is true that name designation for virus as well as for disease is fluid in this fast-moving research field.  We have updated these names in reflecting the advancements.  Even more than this, we proposed to change the disease caused by SARS-CoV-2 from “COVID-19” to “SARS-2” so that the virus and the disease are consistently named and both SARS epidemics can be logically and easily discussed together. (Page 1,2, 36-63)

Point 3

Introduction should be considerably enriched.

Response 3:

The introduction is completely re-written.  It is more comprehensive in coverage but very concise in language. (Page 1,2, 36-63)

Point 4

Figures are of poor quality and should be provided in a higher resolution format.

Response 4:

We have made some changes in two previous figures and provided them in high-resolution formats.  We also added another important figure. (Page 7-9)

Point 5

The purpose of comparing coronaviruses-outbreak is interesting, I think authors should add the MERS.

Response 5:

Thank you for the positive comment.  We have made this comparison more significant by focusing on two repeated SARS outbreaks.  Per your specific suggestion, MERS was also mentioned, mainly as a comparison with SARS. (Page 2, 45-50., Page 3, 96-100)

Point 6

The type of article is confusing: authors speak of analyses, but the format is a review. Please clarify.  

Response 6:

With a new organization and an expanded scope, this manuscript suits well with a mini-review and also provides some insightful analysis.

Point 7

Concerning the phylogenetic tree analysis, how authors performed it, using which software and method/technique. Data are missing and, as such, not well reproducible. Authors speak only of BLAST but should provide more details.  

Response 7:

Based on your suggestion, we have added detailed information in the Figure 1’s legend.(Page 7)

Point 8

There are other articles and studies which should be quoted.

Response 8:

We have cited as many as appropriate all the new studies relevant to out discussion.

Point 9:

Authors should explain the novelty of their study and how can advance and significantly contribute to the field.

Response 9:

As far as we know, this is the first time that current epidemic, known as “Wuhan pneumonia”, “novel coronavirus pneumonia (NCP)” or “COVID-19”, is named as “SARS-2” and thus being discussed with SARS together as repeated outbreak of a same class of disease. It is also the first time SARS epidemiology has been dissected with comprehensive evaluation of a variety of natural and social factors.  We discovered some common environmental conditions conducive for the outbreak of SARS and discussed potential factors shaping the progression and trajectory of SARS.  For better understanding the true origin(s) of SARS-CoV-2 and how the outbreak of SARS-2 happened we present various scenarios so that different clues can be followed to reach the final conclusion.  Regardless of the outcome of this search for truth, the framework set out in this mini-review can provide useful guideline for timely controlling current and effective prevention of future SARS.

Round 2

Reviewer 4 Report

The paper has been significantly improved.